# Bi-Drop: Enhancing Fine-tuning Generalization via Synchronous sub-net Estimation and Optimization

**Shoujie Tong**[1,2*]    **Heming Xia**[1,2*]    **Damai Dai**[1,4]    **Runxin Xu**[1,4]    **Tianyu Liu**[3]
**Binghuai Lin**[5]    **Yunbo Cao**[5]    **Zhifang Sui**[1,4]

[1] National Key Laboratory for Multimedia Information Processing, Peking University
[2] School of Software & Microelectronics, Peking University    [3] Alibaba Group
[4] School of Computer Science, Peking University    [5] Tencent Cloud AI

{tong}@stu.pku.edu.cn {xiaheming,daidamai,tianyu0421,szf}@pku.edu.cn
{runxinxu}@gmail.com {binghuailin, yunbocao}@tencent.com

## Abstract

Pretrained language models have achieved remarkable success in natural language understanding. However, fine-tuning pretrained models on limited training data tends to overfit and thus diminish performance. This paper presents Bi-Drop, a fine-tuning strategy that selectively updates model parameters using gradients from various sub-nets dynamically generated by dropout. The sub-net estimation of Bi-Drop is performed in an in-batch manner, so it overcomes the problem of hysteresis in sub-net updating, which is possessed by previous methods that perform asynchronous sub-net estimation. Also, Bi-Drop needs only one mini-batch to estimate the sub-net so it achieves higher utility of training data. Experiments on the GLUE benchmark demonstrate that Bi-Drop consistently outperforms previous fine-tuning methods. Furthermore, empirical results also show that Bi-Drop exhibits excellent generalization ability and robustness for domain transfer, data imbalance, and low-resource scenarios.

## 1 Introduction

In recent years, Natural Language Processing (NLP) has achieved significant progress due to the emergence of large-scale Pretrained Language Models (PLMs) (Devlin et al., 2018; Liu et al., 2019; Raffel et al., 2019; Clark et al., 2020). For downstream tasks, compared with training from scratch, fine-tuning pretrained models can usually achieve efficient adaptation and result in better performance. Despite the great success, fine-tuning methods still face challenges in maintaining generalization performance on downstream tasks - they tend to run into the overfitting issue when the training data is limited (Phang et al., 2018; Devlin et al., 2018; Lee et al., 2020).

To improve the generalization ability of fine-tuning methods, many regularization techniques have been proposed (Chen et al., 2020; Aghajanyan

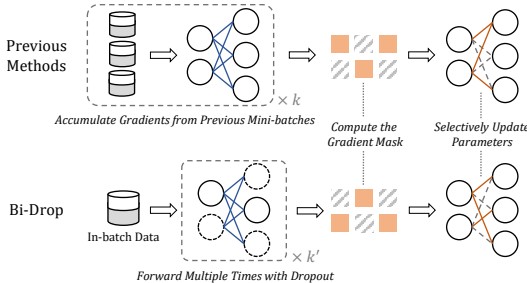

Figure 1: Comparisons between Bi-Drop and previous methods. Unlike previous methods that require multiple mini-batches of data to asynchronously determine the sub-net to optimize, Bi-Drop has a synchronous sub-net selection strategy with a higher data utility.

et al., 2021; Wu et al., 2021; Xu et al., 2021; Yuan et al., 2022), such as sub-net optimization strategies like Child-Tuning$_D$ (Xu et al., 2021) and DPS (Zhang et al., 2022). Child-Tuning$_D$ selects a static sub-net for updating based on parameter importance estimated by Fisher Information (FI). As an improved variant of Child-Tuning$_D$, DPS dynamically decides the sub-net to be updated by estimating FI with multiple mini-batches of data. Although these FI-based methods achieve better generalization ability than vanilla fine-tuning, they still have two limitations: **(1) hysteresis in sub-net updating:** the sub-net preference is estimated with the model parameters in previous iterations and may be incompatible with the current update step; and **(2) insufficient utility of training data:** FI estimation requires cumulative gradients through multiple mini-batches, so these methods cannot fit in situations with data scarcity.

In this paper, we delve deeper into adaptive sub-net optimization strategies and propose Bi-Drop, a *FI-free* strategy for fine-tuning pretrained language models. Unlike Fisher information estimation, which requires cumulative gradients of mini-batches, Bi-Drop only relies on information in a single mini-batch to select the parameters to update. Specifically, Bi-Drop utilizes gradient information

---

*Equal contributions

from different sub-nets dynamically generated by dropout in each mini-batch. As illustrated in Figure 1, within a single training step of Bi-Drop, a mini-batch will go through the forward pass multiple times and, due to the randomness introduced by dropout, yield various distinct sub-nets. We then apply a parameter selection algorithm with perturbation and scaling factors to stabilize the gradient updates. With this synchronous parameter selection strategy, Bi-Drop can selectively update model parameters according to the information from only the current mini-batch, and thus mitigate overfitting with a high utility of training data.

Extensive experiments on the GLUE benchmark demonstrate that Bi-Drop shows remarkable superiority over the state-of-the-art fine-tuning regularization methods, with a considerable margin of $0.53 \sim 1.50$ average score. Moreover, Bi-Drop consistently outperforms vanilla fine-tuning by $0.83 \sim 1.58$ average score across various PLMs. Further analysis indicates that Bi-Drop attains superb generalization ability for domain transfer and task transfer, and is robust for data imbalance and low-resource scenarios.

To sum up, our contributions are three-fold:

- We propose Bi-Drop, a step-wise sub-net optimization strategy that adaptively selects the updated sub-net based on the current mini-batch data. Compared with prior FI-based methods, Bi-Drop derives a more stable and robust training trajectory with simultaneous sub-net update and high utility of training data.[1]
- With extensive experiments on various PLMs, we demonstrate that Bi-Drop achieves consistent and remarkable superiority over prior fine-tuning regularization methods.
- Further analysis shows that Bi-drop attains superb generalization ability for domain transfer and task transfer, and is robust for data imbalance and low-resource scenarios.

## 2 Related Work

**Pretrained Language Models** In recent years, the field of natural language processing (NLP) has witnessed significant advancements due to the development of large-scale pretrained language models (PLMs). The introduction of BERT (Devlin

---

[1]Our code is available at https://github.com/tongshoujie/Bi-Drop

| Models | Sub-net Granularity | Hysteresis | Data Depend. |
|---|---|---|---|
| Child-Tuning$_D$ | *task-wise* | Yes | Yes |
| DPS | *cycle-wise* | Yes | Yes |
| Bi-Drop | *step-wise* | **No** | **No** |

Table 1: Comparison of Bi-Drop with prior FI-based sub-net optimization methods. "cycle-wise" refers to the granularity that includes multiple mini-batches of data.

et al., 2018) sparked a continuous emergence of various pre-trained models, including RoBERTa (Liu et al., 2019), ELECTRA (Clark et al., 2020), XLNet (Yang et al., 2019), GPT-2 (Radford et al., 2019), and GPT-3 (Brown et al., 2020), which have brought remarkable improvements in model structures and scales. Until now, fine-tuning is still one of the most popular approaches to adapting large pretrained language models to downstream tasks.

**Regularization Methods for Fine-tuning** Large-scale PLMs are prone to over-fitting (Phang et al., 2018; Devlin et al., 2018) and exhibit inadequate generalization ability when fine-tuned with limited training data (Aghajanyan et al., 2021; Mahabadi et al., 2021), resulting in degraded performance. To tackle this issue, various regularization techniques have been suggested to enhance the generalization capacity of models, including advanced dropout alternatives (Wan et al., 2013; Wu et al., 2021), applying adversarial perturbations (Aghajanyan et al., 2021; Wu et al., 2022; Yuan et al., 2022) and constrained regularization methods (DauméIII, 2007; Chen et al., 2020). In recent years, Child-tuning (Xu et al., 2021) and DPS (Zhang et al., 2022) propose to estimate parameter importance based on Fisher Information (FI) and selectively optimize a sub-net during fine-tuning to mitigate overfitting.

FI-based methods have a strong dependence on the training data and exhibit hysteresis in sub-net updating. As shown in Table 1, compared with prior FI-based methods, Bi-Drop introduces a step-wise sub-net optimization strategy that adaptively selects the sub-net to be updated based on the current mini-batch. It is worth noting that, as a model-agnostic technique, Bi-Drop is orthogonal to most previous fine-tuning methods, which could further boost the model's performance.

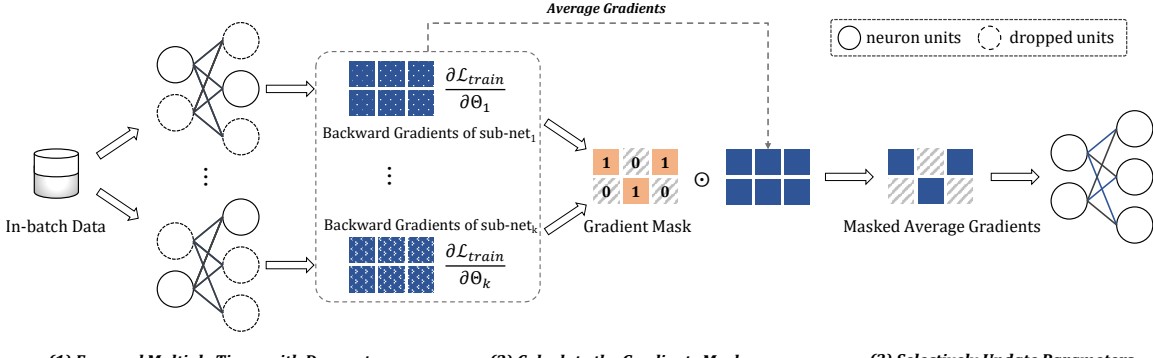

**(1) Forward Multiple Times with Dropout**    **(2) Calculate the Gradients Mask**    **(3) Selectively Update Parameters**

Figure 2: An overall illustration of Bi-Drop. Bi-Drop splits each training step into three sub-steps: **(1) Multiple forward propagations:** each mini-batch sample goes through the forward pass multiple times (denoted as $k$) with dropout; **(2) sub-net selection:** an advanced strategy is adopted to select the sub-net to be updated based on the gradients of distinct sub-nets generated by dropout; **(3) Parameter updating:** only the parameters of the selected sub-net are updated to mitigate overfitting.

## 3 Methodology

### 3.1 Background

We first introduce the paradigm of sub-net optimization by giving general formulations of the backpropagation during the vanilla fine-tuning and CHILD-TUNING$_D$. We denote the parameters of the model at $t$-th iteration as $\boldsymbol{\theta}_t = \{\theta_{t,i}\}_{i=1}^n$, where $\theta_{t,i}$ represent the $i$-th element of $\boldsymbol{\theta}_t$ at the $t$-th training iteration. $\boldsymbol{\theta}_0$ denotes the parameter matrix of the pre-trained model. The vanilla fine-tuning adopts Stochastic Gradient Descent (SGD) to all the model parameters, formally:

$$\boldsymbol{\theta}_{t+1} = \boldsymbol{\theta}_t - \eta \frac{\partial \mathcal{L}(\boldsymbol{\theta}_t)}{\partial \boldsymbol{\theta}_t} \quad (1)$$

where $\mathcal{L}$ represents the training loss within a batch; $\eta$ is the learning rate. Instead of fine-tuning the entire network, CHILD-TUNING$_D$ proposes to only optimize a subset of parameters (i.e., the sub-net). It first adopts the Fisher Information (FI) to estimate the relative importance of the parameters for a specific downstream task, which can be formulated as:

$$F(\boldsymbol{\theta}_0) = \sum_{j=1}^{|\mathcal{D}|} \left( \frac{\partial \mathcal{L}(\boldsymbol{\theta}_0)}{\partial \boldsymbol{\theta}_0} \right)^2 \quad (2)$$

$$M_{CT_D} = F(\boldsymbol{\theta}_0) > \text{sort}\left(F(\boldsymbol{\theta}_0)\right)_p \quad (3)$$

where $\mathcal{D}$ is the training data, $F(\boldsymbol{\theta}_0)$ denotes the fisher information matrix of the pretrained parameters; $\text{sort}(\cdot)_p$ represents the highest value of $p$ percentile in $F(\boldsymbol{\theta}_0)$ after sorting in ascending order; $M_{CT_D}$ is a mask matrix that is the same-sized as $\boldsymbol{\theta}_0$. During fine-tuning, CHILD-TUNING$_D$ only optimizes the selected sub-net in $M_{CT_D}$:

$$\boldsymbol{\theta}_{t+1} = \boldsymbol{\theta}_t - \eta \frac{\partial \mathcal{L}(\boldsymbol{\theta}_t)}{\partial \boldsymbol{\theta}_t} M_{CT_D} \quad (4)$$

### 3.2 Bi-Drop

As introduced in Section 3.1, CHILD-TUNING$_D$ only optimizes an unchanged sub-net during fine-tuning and ignores the update of other parameters, which may degrade the model's performance on downstream tasks. In this section, we offer a detailed introduction to our proposed method, Bi-Drop, which selects the updated parameters adaptively at each fine-tuning step. Specifically, Bi-Drop splits each training step into three sub-steps: (1) multiple forward propagations, (2) sub-net selection, and (3) parameter updating. We provide a pseudo-code of Bi-Drop in Algorithm 1.

#### 3.2.1 Multiple Forward Propagations

Instead of prior FI-based methods that require accumulated gradients to measure the parameter importance, Bi-Drop leverages distinct sub-nets generated by dropout to select the sub-net to be updated. Inspired by Wu et al. (2021), given the training data $\mathcal{D} = \{(x_i, y_i)\}_{i=1}^m$, at each training step, we feed $x_i$ to the model multiple times in the forward pass with different dropouts, and obtain their gradients correspondingly:

$$\mathbf{g}_t^{(j)} = \frac{\partial \mathcal{L}(\boldsymbol{\theta}_t^{(j)})}{\partial \boldsymbol{\theta}_t^{(j)}}, \quad j = 1, 2, ..., k \quad (5)$$

where $\boldsymbol{\theta}_t^{(j)}$ and $\mathbf{g}_t^{(j)}$ represents the parameters of the $j$-th forward pass and its corresponding gradients. $k$ denotes the number of forward passes, i.e., the number of distinct sub-nets with different dropouts.

**Algorithm 1** Bi-Drop for Adam Optimizer

---

**Require:** $\boldsymbol{\theta}_0$: initial pretrained weights; $\mathcal{L}(\boldsymbol{\theta})$: stochastic objective function with parameters $\boldsymbol{\theta}$; $\beta_1, \beta_2 \in [0, 1)$: exponential decay rates for the moment estimates; $\eta$: learning rate;

1: **initialize** timestep $t \leftarrow 0$, first moment vector $\mathbf{m}_0 \leftarrow \mathbf{0}$, second moment vector $\mathbf{v}_0 \leftarrow \mathbf{0}$
2: **while** not converged **do**
3:     $t \leftarrow t + 1$
    *// Multiple forward propagations*
4:     $\mathbf{g}_t^{(j)} \leftarrow \frac{\partial \mathcal{L}(\boldsymbol{\theta}_t^{(j)})}{\partial \boldsymbol{\theta}_t^{(j)}}, \quad j = 1, 2, ..., k$
    *// Sub-net selection*
5:     $M_t \leftarrow \text{SelectSubNetwork}(\mathbf{g}_t)$
    *// Gradients Updating*
6:     $\mathbf{g}_t \leftarrow \mathbf{g}_t \odot M_t$
7:     $\mathbf{m}_t \leftarrow \beta_1 \cdot \mathbf{m}_{t-1} + (1 - \beta_1) \cdot \mathbf{g}_t$
8:     $\mathbf{v}_t \leftarrow \beta_2 \cdot \mathbf{v}_{t-1} + (1 - \beta_2) \cdot \mathbf{g}_t^2$
9:     $\hat{\mathbf{m}}_t \leftarrow \mathbf{m}_t / (1 - \beta_1^t)$
10:     $\hat{\mathbf{v}}_t \leftarrow \mathbf{v}_t / (1 - \beta_2^t)$
    *// Update weights*
11:     $\mathbf{w}_t \leftarrow \mathbf{w}_{t-1} - \eta \cdot \hat{\mathbf{m}}_t / (\sqrt{\hat{\mathbf{v}}_t} + \epsilon)$
12: **end while**
13: **return** $\mathbf{w}_t$

---

### 3.2.2 Sub-net Selection

In this subsection, we introduce our sub-net selection strategy, which estimates the relevant importance of parameters based on the gradients of distinct sub-nets generated by different dropouts. Concretely, our strategy is based on two estimation factors: the perturbation factor and the scaling factor.

**Perturbation Factor** We propose the perturbation factor, which estimates the importance of parameters according to their stability with different dropouts in the forward pass. We point out that various sub-nets generated by dropout can be viewed as adversarial perturbations to the vanilla model. The perturbation factor is formalized as follows:

$$\boldsymbol{\mu}_t = \frac{1}{k} \sum_{j=1}^{k} \mathbf{g}_t^{(j)} \tag{6}$$

$$\mathbf{F}_{\text{per}}(\boldsymbol{\theta}_t) = |\boldsymbol{\mu}_t| \cdot \left[ \sum_j \left( \mathbf{g}_t^{(j)} - \boldsymbol{\mu}_t \right)^2 \right]^{-\frac{1}{2}} \tag{7}$$

where $\boldsymbol{\mu}_t$ is the average gradients of parameters. $\mathbf{F}_{\text{per}}$ measures the stability of parameters by both considering the mean and variance of gradients with adversarial perturbations, i.e. sub-nets with

consistently larger gradients and smaller variances are more favorable by this factor.

**Scaling Factor** We further propose the scaling factor as a regularization term. This factor measures the ratio of the average parameter gradients to the original parameters. Parameters whose gradient scale is much smaller than the original parameters will not be updated, which is similar in spirit to gradient clipping.

$$\mathbf{F}_{\text{sca}}(\boldsymbol{\theta}_t) = |\boldsymbol{\mu}_t| \cdot |\boldsymbol{\theta}_t|^{-1} \tag{8}$$

### 3.2.3 Parameter Updating

Following prior work (Xu et al., 2021; Zhang et al., 2022), we derive a step-wise mask matrix $M_t$ filtered by selecting the highest value of $p$ percentile measured by the aforementioned two estimation factors.

$$\mathbf{F}_{\text{final}}(\boldsymbol{\theta}_t) = \mathbf{F}_{\text{per}}(\boldsymbol{\theta}_t) \cdot \mathbf{F}_{\text{sca}}(\boldsymbol{\theta}_t) \tag{9}$$

$$M_t = \mathbf{F}_{\text{final}}(\boldsymbol{\theta}_t) > \text{sort}\left(\mathbf{F}_{\text{final}}(\boldsymbol{\theta}_t)\right)_p \tag{10}$$

Then, we utilize $M_t$ to update the sub-net which consists of important parameters at each training step. We denote the formulation by simply replacing Eq.4 with our step-wise mask matrix $M_t$:

$$\boldsymbol{\theta}_{t+1} = \boldsymbol{\theta}_t - \eta \boldsymbol{\mu}_t \cdot M_t \tag{11}$$

## 4 Experiments

### 4.1 Datasets

**GLUE Benchmark** Following previous work (Lee et al., 2020; Dodge et al., 2020; Zhang et al., 2021), we conduct extensive experiments on the GLUE benchmark, including natural language inference (RTE, QNLI, MNLI), paraphrase and similarity (MRPC, STS-B, QQP), linguistic acceptability (CoLA), and sentiment classification (SST-2). We include the detailed statistics and metrics of the datasets in Appendix A. Since the test results are only accessible online with limited submission times, we follow prior studies (Phang et al., 2018; Lee et al., 2020; Aghajanyan et al., 2021; Dodge et al., 2020; Xu et al., 2021; Zhang et al., 2022) that fine-tune the pretrained model on the training set and report the results on the development sets using the last checkpoint.

| Method | CoLA | MRPC | RTE | STS-B | Avg | Δ |
|---|---|---|---|---|---|---|
| Vanilla | 63.76(65.55) | 90.41(91.91) | 70.97(74.01) | 89.70(90.58) | 78.71(80.51) | 0.00 |
| Weight Decay | 63.70(65.53) | 90.89(91.98) | 72.24(74.37) | 89.66(90.22) | 79.12(80.53) | +0.41(0.02) |
| Top-K Tuning | 63.89(65.37) | 91.09(92.05) | 72.84(75.17) | 89.64(90.83) | 79.37(80.86) | +0.66(0.35) |
| Mixout | 64.05(65.41) | 91.19(92.05) | 72.32(75.52) | 89.89(90.42) | 79.36(80.85) | +0.65(0.34) |
| RecAdam | 64.09(65.35) | 90.88(91.89) | 72.48(74.30) | 89.67(90.62) | 79.28(80.54) | +0.57(0.03) |
| ChildTuning$_D$ | 64.25(65.81) | 91.19(92.20) | 72.35(76.53) | 90.18(90.88) | 79.49(81.36) | +0.78(0.85) |
| R3F | 64.88(66.33) | 91.55(92.28) | 72.42(74.37) | 89.70(90.58)* | 79.64(80.89) | +0.93(0.38) |
| DPS$_{mix}$ | 64.40(66.21) | 91.05(92.17) | 72.88(75.81) | 90.47(91.02) | 79.70(81.30) | +0.99(0.79) |
| R-Drop | 64.83(**66.79**) | 91.76(92.17) | 73.12(76.53) | 90.14(90.42) | 79.96(81.48) | +1.25(0.97) |
| Bi-Drop | **64.94**(66.69) | **91.79(92.68)** | **73.79(77.61)** | **90.59(91.04)** | **80.26(82.01)** | **+1.55(1.50)** |

Table 2: Comparison between Bi-Drop with prior fine-tuning methods. We report the mean (max) results of 10 random seeds. The best results are **bold**. Note that since R3F is not applicable to regression, the result on STS-B (marked with *) remains the same as vanilla. Bi-Drop achieves the best performance compared with other methods.

**NLI Datasets** We also evaluate the generalization ability of Bi-Drop on several Natural Language Inference (NLI) tasks, including SNLI (Bowman et al., 2015), MNLI (Williams et al., 2018), MNLI-M (Williams et al., 2018) and SICK (Marelli et al., 2014). We report all results by Accuracy on the development sets consistent with GLUE.

## 4.2 Baselines

Besides the vanilla fine-tuning method, we mainly compare Bi-Drop with the following baselines:

**Mixout** (Lee et al., 2020) is a fine-tuning technique that stochastically replaces the parameters with their pretrained weight based on the Bernoulli distribution. **R3F** (Aghajanyan et al., 2021) is a fine-tuning strategy motivated by trust-region theory, which injects noise sampled from either a normal or uniform distribution into the pre-trained representations. **R-Drop** (Wu et al., 2021) minimizes the bidirectional KL-divergence to force the output distributions of two sub-nets sampled by dropout to be consistent with each other. **Child-Tuning$_D$** (Xu et al., 2021) selects the task-relevant parameters as the sub-net based on the Fisher information and only updates the sub-net during fine-tuning. **DPS** (Zhang et al., 2022) is a dynamic sub-net optimization algorithm based on Child-Tuning$_D$. It estimates Fisher information with multiple minibatches of data and selects the sub-net adaptively during fine-tuning.

For reference, we also show other prior finetuning techniques in our main experimental results, such as Weight Decay (DauméIII, 2007), Top-K Tuning (Houlsby et al., 2019) and RecAdam (Chen et al., 2020).

## 4.3 Experiments Setup

We conduct our experiments based on the HuggingFace transformers library [2] (Wolf et al., 2020) and follow the default hyper-parameters and settings unless noted otherwise. We report the averaged results over 10 random seeds. Other detailed experimental setups are presented in Appendix B.

## 4.4 Results on GLUE

**Comparison with Prior Methods** We compare Bi-Drop with various prior fine-tuning methods based on BERT$_{large}$ and report the mean (and max) scores on GLUE benchmark in Table 2, following Lee et al. (2020) and Xu et al. (2021). The results indicate that Bi-Drop yields the best average performance across all tasks, showing its effectiveness. Moreover, the average of the maximum scores attained by Bi-Drop is superior to that of other methods, providing further evidence of the effectiveness of Bi-Drop. We also conducted the same experiment on Roberta$_{large}$, and the details can be found in Appendix E.

**Comparison with Vanilla Fine-tuning** We show the experimental results of six widely used largescale PLMs on the GLUE Benchmark in Table 3. The results show that Bi-Drop outperforms vanilla fine-tuning consistently and significantly across all tasks performed on various PLMs. For instance, Bi-Drop achieves an improvement of up to 1.58 average score on BERT$_{base}$ and 1.35 average score on Roberta$_{base}$. The results highlight the universal effectiveness of Bi-Drop in enhancing the fine-tuning performance of PLMs. Additionally, because Bi-Drop forward-propagate twice, we present an additional study of the baseline with doubled batch

---

[2] https://github.com/huggingface/transformers

| Method | BERT$_{base}$ | | | | | Roberta$_{base}$ | | | | |
|---|---|---|---|---|---|---|---|---|---|---|
| | CoLA | MRPC | RTE | STS-B | Avg | CoLA | MRPC | RTE | STS-B | Avg |
| Vanilla | 57.67 | 90.38 | 69.57 | 89.35 | 76.74 | 59.45 | 91.94 | 76.28 | 90.60 | 79.57 |
| Bi-Drop | **60.76** | **91.01** | **71.73** | **89.78** | **78.32** | **61.28** | **92.40** | **78.99** | **91.00** | **80.92** |

| Method | BERT$_{large}$ | | | | | Roberta$_{large}$ | | | | |
|---|---|---|---|---|---|---|---|---|---|---|
| | CoLA | MRPC | RTE | STS-B | Avg | CoLA | MRPC | RTE | STS-B | Avg |
| Vanilla | 63.76 | 90.41 | 70.97 | 89.70 | 78.71 | 66.01 | 92.56 | 84.51 | 92.05 | 83.78 |
| Bi-Drop | **64.94** | **91.79** | **73.79** | **90.50** | **80.26** | **68.03** | **92.95** | **86.10** | **92.36** | **84.86** |

| Method | DeBERTa$_{large}$ | | | | | ELECTRA$_{large}$ | | | | |
|---|---|---|---|---|---|---|---|---|---|---|
| | CoLA | MRPC | RTE | STS-B | Avg | CoLA | MRPC | RTE | STS-B | Avg |
| Vanilla | 65.18 | 92.32 | 85.60 | 91.64 | 83.69 | 70.02 | 92.94 | 88.37 | 91.91 | 85.81 |
| Bi-Drop | **66.91** | **92.88** | **86.72** | **92.33** | **84.71** | **71.29** | **93.68** | **88.98** | **92.61** | **86.64** |

Table 3: Comparison between Bi-Drop and vanilla fine-tuning applied to six widely-used large-scale PLMs. We report the mean results of 10 random seeds. Average scores on all tasks are underlined. The best results are **bold**. It shows that Bi-Drop yields consistent improvements across all tasks among different PLMs.

| Datasets | SNLI | | | | | MNLI | | | | |
|---|---|---|---|---|---|---|---|---|---|---|
| | Vanilla | CT$_D$ | R-Drop | DPS | Bi-Drop | Vanilla | CT$_D$ | R-Drop | DPS | Bi-Drop |
| MNLI | 64.53 | 64.16 | 64.51 | 64.70 | **66.88** | 75.78 | 76.27 | **76.85** | 75.48 | 76.63 |
| MNLI–m | 66.11 | 66.30 | 66.59 | 67.29 | **68.49** | 77.31 | 77.30 | 77.93 | 77.45 | **77.98** |
| SNLI | 83.37 | 83.53 | 83.59 | 82.95 | **83.62** | 70.87 | 70.76 | 71.48 | 71.56 | **71.63** |
| SICK | 52.59 | 53.73 | 53.59 | **55.89** | 54.28 | 53.27 | **55.35** | 54.21 | 54.81 | 54.93 |
| **Avg** | 61.08 | 61.40 | 61.56 | 62.63 | **63.22** | 67.15 | 67.80 | 67.87 | 67.94 | **68.18** |
| $\Delta_{avg}$ | – | ↑ 0.32 | ↑ 0.48 | ↑ 1.55 | ↑ **2.14** | – | ↑ 0.65 | ↑ 0.72 | ↑ 0.79 | ↑ **1.03** |

Table 4: **Evaluation for out-of-domain generalization**. The models are trained on MNLI/SNLI and tested on out-of-domain data. Average scores are computed excluding in-domain results (underlined). The best results are **bold**. Bi-Drop can better maintain the out-of-domain generalization ability of the model.

size in Appendix D.

## 4.5 Out-of-Domain Generalization

We further evaluate the generalization ability of Bi-Drop on a widely used experimental setting in prior research (Aghajanyan et al., 2021; Xu et al., 2021; Zhang et al., 2022): out-of-domain generalization. In detail, we finetune BERT$_{large}$ with different strategies on $5k$ subsampled MNLI and SNLI datasets respectively, and directly evaluate the fine-tuned model on other NLI datasets. The experimental results in Table 4 illustrate that Bi-Drop outperforms vanilla fine-tuning and prior methods across various datasets. Specifically, compared with other fine-tuning methods on SNLI, Bi-Drop demonstrates consistent and substantial improvement, with an average score increase of 2.14. In particular, Bi-Drop achieves an improvement of 2.35 on MNLI task and 2.38 on MNLI-m task. For models trained on MNLI, Bi-Drop also consistently outperforms prior methods, with an average improvement of 1.03 score. The experimental results

indicate that Bi-Drop encourages the model to learn deeper and more generic semantic features and alleviate superficial biases of specific tasks, which improves the model's generalization ability.

## 4.6 Task Generalization

We also evaluate the generalization ability of fine-tuned models following the experimental setting of Aghajanyan et al. (2021) and Xu et al. (2021), which freezes the representations of the model fine-tuned on one task and only trains the linear classifier on the other task. Specifically, we finetune BERT$_{large}$ among one task selected among MRPC, CoLA, and RTE and then transfer the model to the other two tasks. Figure 3 shows that Bi-Drop consistently outperforms vanilla fine-tuning when the fine-tuned model is transferred to other tasks. In particular, Bi-Drop improves by 3.50 and 3.28, when models trained on MRPC and RTE respectively are evaluated on CoLA. The results further verify the conclusion that Bi-Drop helps models learn more generalizable representations, com-

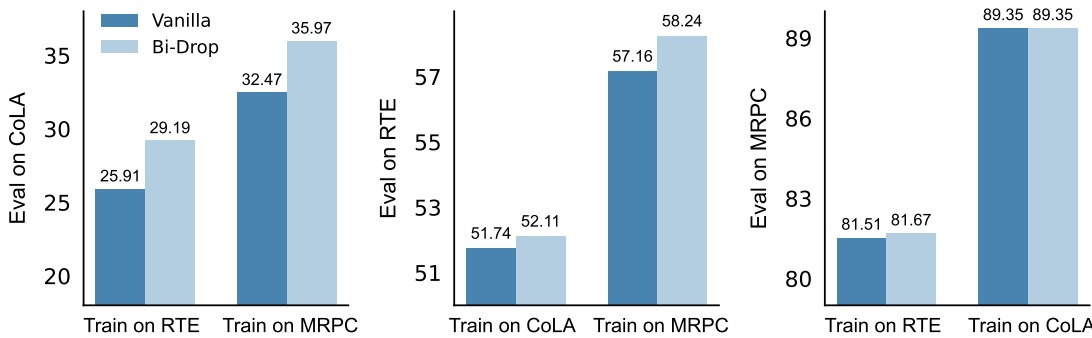

Figure 3: **Evaluation for task generalization**. The model is fine-tuned on a specific task among MRPC, CoLA, RTE and transferred to the other two tasks. Bi-Drop can be more generalizable.

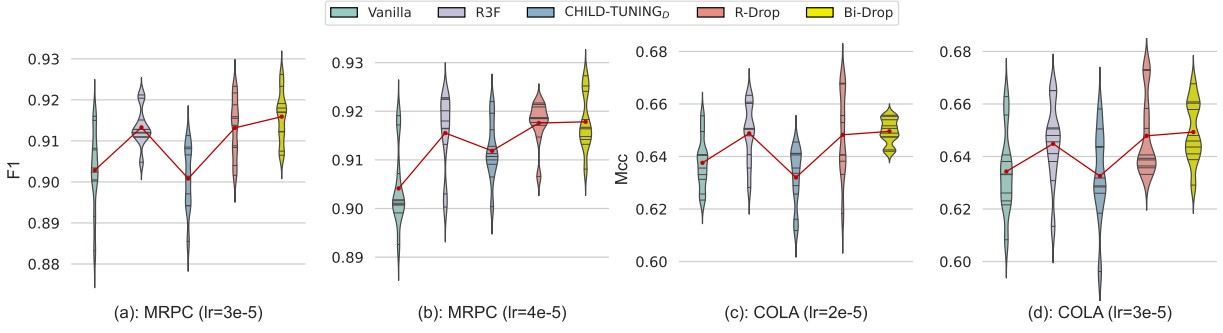

Figure 4: Results of Bi-Drop across four experimental settings. Each method includes a violin plot for 10 random runs. Compared with other methods, the shorter and thicker violin plot of Bi-Drop proves its better stability.

pared with the vanilla fine-tuning approach.

## 5   Analysis

### 5.1   Stability to Random Seeds

We further investigate the stability properties of fine-tuned models. Figure 4 shows the output distributions of models with four experimental settings and across 10 random seeds. The results demonstrate that Bi-Drop outperforms other strategies in terms of average performance, and also exhibits greater stability by achieving more consistent results across 10 random seeds with lower variance.

### 5.2   Robustness Analysis

Recent research has brought to light that the vanilla fine-tuning approach is prone to deception and vulnerability in many aspects. In this study, we assess the robustness of Bi-Drop by designing evaluation tasks that focus on two common scenarios, aiming to examine its ability to withstand various forms of perturbations while maintaining its robustness.

**Robustness to Label Noise**   Due to the inherent limitations of human annotation, widely-used large-scale datasets inevitably contain a certain amount of incorrect labels (Vasudevan et al., 2022). To

investigate the robustness of Bi-Drop to label noise, we conduct simple simulation experiments on RTE, MRPC, and CoLA by randomly corrupting a pre-determined fraction of labels with erroneous values. We evaluate the robustness of various fine-tuning methods trained on noisy data. The results shown in the left panel of Table 5 demonstrate that Bi-Drop achieves consistent superiority to other fine-tuning methods on noisy data. Furthermore, we conducted a comparison of model performance degradation across varying noise ratios. We calculated the degradation and presented it in brackets, as compared with Table 2. It indicates that Bi-Drop has the smallest performance drop compared with other fine-tuning methods. These results collectively demonstrate that Bi-Drop is more robust to label noise than prior methods.

**Robustness to Data Imbalance**   Minority class refers to the class that owns insufficient instances in the training set. In this section, we strive to explore the robustness of diverse fine-tuning approaches for the minority class by carrying out experiments on synthetic RTE, MRPC, and CoLA datasets. The experimental results are illustrated in the right panel of Table 5, which shows that Bi-Drop significantly outperforms other fine-tuning methods. Bi-Drop

**Table 5 Left: Robustness to label noise**

| Method | CoLA | MRPC | RTE | Avg |
|---|---|---|---|---|
| | Noise Ratio: 5% | | | |
| vanilla | 61.39 | 90.24 | 69.78 | 73.80 ($\downarrow$ 1.25) |
| $CT_D$ | 61.66 | 90.39 | 71.47 | 74.51 ($\downarrow$ 1.77) |
| R-Drop | 62.33 | 91.21 | 73.16 | 75.57 ($\downarrow$ 1.00) |
| Bi-Drop | 62.29 | **91.85** | **73.57** | **75.90** ($\downarrow$ **0.94**) |
| | Noise Ratio: 10% | | | |
| vanilla | 59.35 | 89.78 | 68.51 | 72.55 ($\downarrow$ 2.50) |
| $CT_D$ | 61.56 | 89.88 | 69.78 | 73.72 ($\downarrow$ 2.56) |
| R-Drop | 61.12 | 91.10 | 71.36 | 74.53 ($\downarrow$ 2.04) |
| Bi-Drop | **61.65** | 91.12 | **71.96** | **74.91** ($\downarrow$ **1.93**) |
| | Noise Ratio: 15% | | | |
| vanilla | 58.96 | 88.10 | 68.14 | 71.73 ($\downarrow$ 3.32) |
| $CT_D$ | 59.99 | 88.10 | 68.88 | 72.32 ($\downarrow$ 3.96) |
| R-Drop | **60.80** | 89.26 | 69.55 | 73.20 ($\downarrow$ 3.36) |
| Bi-Drop | 60.08 | **89.73** | **70.88** | **73.56** ($\downarrow$ **3.28**) |

**Table 5 Right: Robustness to data imbalance**

| Method | CoLA | MRPC | RTE | Avg |
|---|---|---|---|---|
| | Reduction Ratio: 70% | | | |
| Vanilla | 81.12 | 78.88 | 27.71 | 62.57 ( – ) |
| $CT_D$ | **83.10** | 79.50 | 28.14 | 63.58 ($\uparrow$ 1.01) |
| R-Drop | 81.37 | 76.46 | 32.61 | 63.48 ($\uparrow$ 0.91) |
| Bi-Drop | 81.88 | **80.17** | **37.66** | **66.57** ($\uparrow$ **4.00**) |
| | Reduction Ratio: 60% | | | |
| Vanilla | 83.30 | 83.98 | 39.32 | 68.87 ( – ) |
| $CT_D$ | **85.78** | 84.55 | 37.65 | 69.33 ($\uparrow$ 0.46) |
| R-Drop | 83.55 | 85.07 | 43.51 | 70.71 ($\uparrow$ 1.84) |
| Bi-Drop | 84.33 | 85.35 | **49.62** | **73.10** ($\uparrow$ **4.23**) |
| | Reduction Ratio: 50% | | | |
| Vanilla | 86.07 | 87.40 | 45.27 | 72.91 ( – ) |
| $CT_D$ | **87.98** | 87.64 | 50.78 | 75.47 ($\uparrow$ 2.56) |
| R-Drop | 86.04 | 88.89 | 55.73 | 76.89 ($\uparrow$ 3.98) |
| Bi-Drop | 86.73 | **89.61** | **58.27** | **78.20** ($\uparrow$ **5.29**) |

Table 5: **Left: Robustness to label noise.** The noise ratio is the percentage of training instances whose labels are transferred to incorrect labels. **Right: Robustness to data imbalance.** We reduce the number of instances labeled 1 by 70%/60%/50% in the training set and test the accuracy of instances labeled 1 (as the minority class) in the validation set. Bi-Drop can maintain more robust representations compared with other fine-tuning methods.

| Datasets | 0.5K | | | | 1K | | | |
|---|---|---|---|---|---|---|---|---|
| | Vanilla | $CT_D$ | DPS | Bi-Drop | Vanilla | $CT_D$ | DPS | Bi-Drop |
| CoLA | 36.23 | 40.77 | 41.87 | **44.78** | 48.92 | 51.63 | 52.89 | **54.76** |
| MRPC | 81.33 | 81.95 | **83.29** | 83.18 | 83.90 | 84.62 | 85.03 | **85.44** |
| RTE | 58.87 | 59.27 | 59.16 | **60.04** | 62.17 | 63.85 | 64.92 | **66.96** |
| STS-B | 82.60 | 82.79 | 83.41 | **85.19** | 85.91 | 86.92 | 87.26 | **87.53** |
| SST-2 | 86.11 | 88.19 | 89.08 | **89.38** | 89.95 | 89.88 | **90.44** | 90.41 |
| QNLI | 78.76 | 79.32 | 79.27 | **80.48** | 82.49 | 83.65 | 83.86 | **84.35** |
| QQP | 71.88 | 73.43 | 74.22 | **74.61** | 77.65 | 78.57 | 78.79 | **79.01** |
| MNLI | 46.44 | 45.74 | 47.63 | **50.79** | 56.55 | 60.96 | 59.47 | **61.25** |
| **Avg** | 67.78 | 68.93 | 69.74 | **71.06** | 73.44 | 75.01 | 75.33 | **76.21** |
| $\Delta_{avg}$ | – | $\uparrow$ 1.15 | $\uparrow$ 1.96 | $\uparrow$ **3.28** | – | $\uparrow$ 1.57 | $\uparrow$ 1.89 | $\uparrow$ **2.77** |

Table 6: Comparison between Bi-Drop and prior sub-net optimization strategies with varying low-resource scenarios (0.5K, 1K). We report the results of 10 random seeds and the best results are **bold**. Bi-Drop performs better than other methods in low-resource scenarios.

achieves up to 4.00, 4.23, and 5.29 average score improvements on 30%, 40%, and 50% reduction ratios respectively, outperforming other fine-tuning methods at lower reduction ratios and showcasing its robustness towards the minority class.

## 5.3 Performance in Low-Resource Scenarios

As illustrated in Section 1 and 2, compared with prior FI-based sub-net optimization methods that have a strong dependence on the training data, Bi-Drop proposes a step-wise sub-net selection strategy, which chooses the optimized parameters with the current mini-batch. In this section, we conduct extensive experiments to analyze how this dependency affects the performance of models. Concretely, we adopt various fine-tuning methods on

BERT$_{large}$ with a limited amount of training data. The results are illustrated in Table 6. As the data amount decreases from 1.0K to 0.5K, the average improvement score of Child-Tuning$_D$ over vanilla fine-tuning decreases from 1.57 to 1.15, while its improved variant DPS maintains a relatively stable improvement. But Bi-Drop improves the average improvement score from 2.77 to 3.28. The results indicate the superiority of Bi-Drop over prior methods in low-resource scenarios.

## 5.4 Ablation Study

To evaluate the effectiveness of our proposed fine-tuning strategy, we conduct an ablation study in Table 7. The results show that both our sub-net selection strategy and gradient averaging strategy

| Method | CoLA | MRPC | RTE | STS-B | Avg | $\Delta$ |
|---|---|---|---|---|---|---|
| Bi-Drop ($g_{avg}$ + ESS) | **64.94** | **91.79** | **73.79** | **90.59** | **80.26** | **+1.55** |
| $g_{avg}$ + Perturbation Factor | 64.71 | 91.67 | 73.21 | 90.36 | 79.99 | +1.28 |
| $g_{avg}$ + Scaling Factor | 64.38 | 91.63 | 72.96 | 90.41 | 79.85 | +1.14 |
| $g_{avg}$ + RSS | 64.24 | 90.96 | 71.73 | 90.25 | 79.30 | +0.59 |
| $g_{avg}$ | 63.82 | 91.26 | 71.16 | 89.81 | 79.01 | +0.30 |
| Vanilla | 63.76 | 90.41 | 70.97 | 89.70 | 78.71 | 0.00 |

Table 7: Ablation results. ESS represents our Effective Sub-net Selection strategy using both factors Perturbation and Scaling. RSS stands for Random Sub-net Selection strategy. Both our sub-net selection strategy and gradient averaging strategy are effective.

contribute to the performance improvement of Bi-Drop.

# 6 Conclusion

In this work, we propose a new sub-net optimization technique for large-scale PLMs, named Bi-Drop, which leverages the gradients of multiple sub-nets generated by dropout to select the updated parameters. Extensive experiments on various downstream tasks demonstrate that Bi-Drop achieves consistent and remarkable improvements over vanilla fine-tuning and prior excellent approaches by a considerable margin, across various model architectures. Further analysis indicates the generalizability and robustness of Bi-Drop over transferring, data imbalance and low-resource experiments.

# 7 Limitations

We propose a novel and effective fine-tuning method, Bi-Drop, which achieves a considerable performance improvement in downstream tasks. However, similar to some previous studies(Jiang et al., 2020; Aghajanyan et al., 2021; Wu et al., 2021), Bi-Drop requires multiple forward propagations, which makes its training time efficiency not good enough compared with the vanilla fine-tuning method.

# Acknowledgements

This paper is supported by the National Key Research and Development Program of China 2020AAA0106700 and NSFC project U19A2065.

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

# Appendix

## A  GLUE Benchmark Datasets

In this paper, we conduct experiments on datasets in GLUE benchmark (Wang et al., 2019). The statistical information of the GLUE benchmark is shown in Table 8.

| Dataset | #Train | #Dev | Metrics |
|---------|--------|------|---------|
| *Single-sentence Tasks* | | | |
| CoLA | 8.5k | 1.0k | Matthews Corr |
| SST-2 | 67k | 872 | Accuracy |
| *Inference* | | | |
| RTE | 2.5k | 277 | Accuracy |
| QNLI | 105k | 5.5k | Accuracy |
| MNLI | 393k | 9.8k | Accuracy |
| *Similarity and Paraphrase* | | | |
| MRPC | 3.7k | 408 | F1 |
| STS-B | 5.7k | 1.5k | Spearman Corr |
| QQP | 364k | 40k | F1 |

Table 8: Statistics and metrics of eight datasets used in this paper form GLUE benchmark.

## B  Experimental Details

In this paper, we fine-tune different large pre-trained language models with Bi-Drop, including $\text{BERT}_{\text{BASE}}$[3], $\text{BERT}_{\text{LARGE}}$[4], $\text{RoBERTa}_{\text{BASE}}$[5], $\text{RoBERTa}_{\text{LARGE}}$[6], $\text{DeDERTa}_{\text{LARGE}}$[7], and $\text{ELECTRA}_{\text{LARGE}}$[8]. The training epochs/steps, batch size, and warmup steps are listed in Table 9.

For the glue dataset, our maximum length is set as 128. We use grid search for learning rate from $\{1\text{e-}5, 2\text{e-}5, \ldots, 1\text{e-}4\}$. For Bi-Drop, we use grid search for dropout rate from $\{0.05, 0.1\}$. The number of forward passes is fixed to two($k = 2$). We conduct all the experiments on a single A40 GPU.

---

[3] https://huggingface.co/bert-base-uncased/tree/main
[4] https://huggingface.co/bert-large-cased/tree/main
[5] https://huggingface.co/roberta-base/tree/main
[6] https://huggingface.co/roberta-large/tree/main
[7] https://huggingface.co/microsoft/deberta-large/tree/main
[8] https://huggingface.co/google/electra-large-discriminator/tree/main

## C  Hyper-Parameter Analysis

Bi-Drop uses two dropout techniques. In order to analyze the impact of the dropout rate on the experimental results, a simple analysis experiment was done here. In order to make the comparison fair, all the parameters except the dropout rate are kept the same in the experiment. For simplicity, the dropout values are the same twice. The experimental results are shown in 5. It can be seen that different datasets have different preferences for dropout values. CoLA and RTE achieve the best results when the dropout value is 0.05; while MRPC achieves the best results when the dropout value is 0.1; STSB is insensitive to the dropout value until the dropout value is less than 0.1.

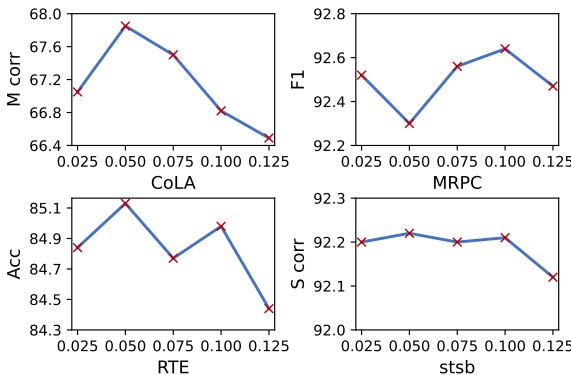

Figure 5: The effect of dropout rate on experimental results

## D  Batch Size Doubled Training

We implement Bi-Drop by repeating the input data twice and forward-propagating twice. This is similar to doubling the batch size at each step. The difference is that half of the data is the same as the other half, and directly doubling the batch size, the data in the same mini-batch is all different. So for a fair comparison, we experimented with directly doubling the batch size. So for a fair comparison, we experimented with directly doubling the batch size. The experimental results are shown in Table 10, results show that directly doubling the batch size has basically no improvement, and Bi-Drop is significantly better than directly doubling the batch size.

## E  Comparison with Prior Methods on Roberta$_{\text{large}}$

We compare Bi-Drop with various prior fine-tuning methods based on $\text{BERT}_{\text{large}}$ and report the mean

| Model | Dataset | Batch Size | Epochs/Steps | Warmup Ratio/Steps |
|---|---|---|---|---|
| BERT | all | 16 | 3 epochs | 10% |
| Roberta | CoLA | 16 | 5336 steps | 320 steps |
| | RTE | 16 | 2036 steps | 122 steps |
| | MRPC | 16 | 2296 steps | 137 steps |
| | STS-B | 16 | 3598 steps | 214steps |
| ELECTRA | CoLA | 32 | 3 epochs | 10% |
| | RTE | 32 | 10 epochs | 10% |
| | MRPC | 32 | 3 epochs | 10% |
| | STS-B | 32 | 10 epochs | 10% |
| DeBERTa | CoLA | 32 | 6 epochs | 100 steps |
| | RTE | 32 | 6 epochs | 50 steps |
| | MRPC | 32 | 6 epochs | 10 steps |
| | STS-B | 32 | 4 epochs | 100 steps |

Table 9: Hyperparameters settings for different pretrained models on variant tasks. These settings are reported in their official repository for *best practice*.

| Method | CoLA | MRPC | RTE | STS-B | Avg |
|---|---|---|---|---|---|
| Vanilla | 57.67 | 90.38 | 69.57 | 89.35 | 76.74 |
| Vanilla(double bsz) | 57.69 | 90.08 | 69.85 | 89.55 | 76.79 |
| Bi-Drop | **60.76** | **91.01** | **71.73** | **89.78** | **78.32** |

Table 10: Comparison of Bi-Drop and directly doubling the batch size. Bi-Drop is significantly better than directly doubling the batch size.

(and max) scores on GLUE benchmark in Table 11,
following Lee et al. (2020) and Xu et al. (2021).

| Method | CoLA | MRPC | RTE | STS-B | Avg | $\Delta$ |
|---|---|---|---|---|---|---|
| Vanilla | 66.01(68.03) | 92.56(93.66) | 84.51(86.28) | 92.05(92.22) | 83.78(85.05) | 0.00 |
| ChildTuning$_D$ | 66.82(68.21) | 92.67(93.58) | 85.89(87.72) | 92.36(92.53) | 84.44(85.51) | +0.66(0.46) |
| DPS$_{mix}$ | 66.86(68.53) | 92.51(93.89) | 85.37(87.72) | **92.47(92.73)** | 84.30(85.72) | +0.52(0.67) |
| R-Drop | 67.26(69.63) | 92.47(93.62) | 85.44(87.63) | 92.42(92.58) | 84.40(85.87) | +0.62(0.82) |
| Bi-Drop | **68.03(70.89)** | **92.95(94.66)** | **86.10(88.09)** | 92.36(92.58) | **84.86(86.56)** | **+1.08(1.51)** |

Table 11: Comparison between Bi-Drop with prior fine-tuning methods. We report the mean (max) results of 10 random seeds. The best results are **bold**. Note that since R3F is not applicable to regression, the result on STS-B (marked with $*$) remains the same as vanilla. Bi-Drop achieves the best performance compared with other methods.