# OpenReview forum: "Bi-Drop: Enhancing Fine-tuning Generalization via Synchronous sub-net Estimation and Optimization"
_EMNLP/2023/Conference — EMNLP 2023 Findings_

### Official Review · Reviewer_NTTU · 2023-07-26

**Typos Grammar Style And Presentation Improvements:** Equation 2 seems not correct in terms…
**Soundness:** 4

**Excitement:**

3: Ambivalent: It has merits (e.g., it reports state-of-the-art results, the idea is nice), but there are key weaknesses (e.g., it describes incremental work), and it can significantly benefit from another round of revision. However, I won't object to accepting it if my co-reviewers champion it.

**Paper Topic And Main Contributions:**

This paper introduces a batch-wise subnetwork detection and parameter update method for the purpose of avoiding overfitting in the course of fine-tuning pretrained language models.  Instead of estimating a static sub-network, this paper explores gradient-related information to figure out important parameters through feeding the current min-batch data several times. The experimental results though not be improved a lot but shows the superiority of the proposed method than previous works.

**Reasons To Accept:**

1.Good and enough experiments, the expiration to OoD generalization and task generalization is interesting. In the scenario of continual pre-training, weights enjoying task generalization would be an indicator for avoiding forgetting. Further exploration in continual pretraining would be a good consequence of the proposed method.
2. The proposed Perturbation Factor and Scaling Factor are effective and straightforward.

**Reasons To Reject:**

1. What is the significant motivation of leveraging stepwise sub-network detection and parameter update? The line of 187 seems the real motivation of the proposed method, compared with previous method of Child-tuningD, but there is no empirical or theoretical proof to support the claim. Intuitively if the static sub-network is estimated with all training data, then it would be good enough. In contrast, the subnetwork is detected and updated in each mini-batch optimization in a dynamic manner, this means the subnetwork can only fit the provided minibatch of data, but never know its generalization to other data in the training set. How randomness introduces benefits should be the strong point of this paper. One potential method is to check if there is strong overlapping among training trajectories in terms of the selected parameters. Compared to child-tuning, the selected sub-network is much smaller for whatever promising reasons. A case is SGD introduces an approximated regularization to the model, contributing to a better solution. This is the theoretical benefits of SGD than GD. I will definitely raise my score if empirical proof is available for why this method is better than child-tuning, speaking of stepwise subnetwork update. Please clearly state the strong point and motivation, uncovering the novelty of this work.
2. The line of 76 is ambiguous and can not explicitly show why reusing training data is helpful for mitigating overfitting. Intuitively reusing training data can accelerate convergence but it is an unanswered question for better generalization. More references or experiments are required for this claim.
3. Also there should be a structural issue in terms of identifying subnetworks. Are the attention weights and weights from other layers considered equally?
4. With the K times feedforward processes, we can estimate the importance of each parameter. Ablation studies w.r.t K should be available. And the batch size and the highest value of p percentile. We need detailed analysis of why stepwise version of subnetwork detection & update, apparently the proposed method is very sensitive to these three factors.

**Reproducibility:**

5: Could easily reproduce the results.

**Reviewer Confidence:**

3: Pretty sure, but there's a chance I missed something. Although I have a good feel for this area in general, I did not carefully check the paper's details, e.g., the math, experimental design, or novelty.

---

> ### Author Rebuttal · Authors · 2023-08-29
>
> ***Q1: What is the significant motivation of leveraging stepwise sub-network detection and parameter update? The line of 187 seems the real motivation of the proposed method, compared with previous method of Child-tuningD, but there is no empirical or theoretical proof to support the claim. Please clearly state the strong point and motivation, uncovering the novelty of this work.***
>
> R1: We appreciate your inquiry of the empirical proof of Bi-Drop. As we illustrate in the paper, Child-tuningD [1] chooses an unchanged subnetwork to optimize and ignores the update of other parameters during the whole finetuning process, which may affect the model’s perception of downstream tasks. In other words, **since there is a misalignment between the original pretrained model and the fine-tuned one, the importance of model parameters may vary from the estimation before** **tuning**. To handle this problem, DPS [2] proposes to dynamically select the optimized sub-network from multiple mini-batches of data. However, these FI-based methods still have two limitations: (1) **hysteresis in sub-net updating**: the sub-net preference is estimated with the model parameters in previous iterations and may be incompatible with the current update step; and (2) **insufficient utility of** **training data**: FI estimation requires cumulative gradients through multiple mini-batches, so these methods cannot fit in situations with data scarcity.
>
> In this work, we propose Bi-Drop, a **novel** ***FI-free*** **method** to estimate the importance of model parameters. Compared with previous methods that choose the optimized sub-net based on the accumulated gradients of multiple mini-batches, Bi-Drop derives a more stable and robust training trajectory with **simultaneous sub-net update** and **high utility of** **training data**. Experimental results illustrated in the paper demonstrate the superiority over prior methods as well as its generalization ability and robustness.
>
> ***Q2: The line of 76 is ambiguous and can not explicitly show why reusing training data is helpful for mitigating overfitting. Intuitively reusing training data can accelerate convergence but it is an unanswered question for better generalization. More references or experiments are required for this claim.***
>
> R2: Here we would like to make clarifications on the claims of the 76 line in our paper. We emphasize the strengths of Bi-Drop with two aspects: Firstly, Bi-Drop proposes a novel training trajectory to selectively update model parameters during fine-tuning, which contributes to better generalization capabilities of models; Secondly, compared with previous methods (such as [1] and [2]) that have high dependences on the data amount, Bi-Drop can easily fit in situations with data scarcity, with a high utility of training data.
>
> ***Q3: Also there should be a structural issue in terms of identifying subnetworks. Are the attention weights and weights from other layers considered equally?***
>
> R3: Following [1] and [2], we treat all parameters in the Transformer Block equally to estimate their importance to the downstream task. However, we appreciate your inquiry of the structural difference in identifying subnetworks, which is an interesting point to explore. We are committed to further exploring this in our ongoing and future research endeavors.
>
> ***Q4: With the K times feedforward processes, we can estimate the importance of each parameter. Ablation studies w.r.t K should be available. And the batch size and the highest value of p percentile. We need detailed analysis of why stepwise version of subnetwork detection & update, apparently the proposed method is very sensitive to these three factors.***
>
> R4: In our main experiments, we set the same batch size as [1] and [2] for fair comparison, which is illustrated in Appendix B. Besides, we have also implemented the baseline with doubling the batch size, to fairly illustrate the superiorty of Bi-Drop. We will include more ablation studies with various k and p percentile in the revised version.
>
> [1] Xu et al. Raise a Child in Large Language Model: Towards Effective and Generalizable Fine-tuning. In EMNLP 2021.
>
> [2] Zhang et al. Fine-Tuning Pre-Trained Language Models Effectively by Optimizing Subnetworks Adaptively. In NIPS 2022.

---

### Official Review · Reviewer_uYzG · 2023-08-04

**Soundness:** 4

**Excitement:**

3: Ambivalent: It has merits (e.g., it reports state-of-the-art results, the idea is nice), but there are key weaknesses (e.g., it describes incremental work), and it can significantly benefit from another round of revision. However, I won't object to accepting it if my co-reviewers champion it.

**Paper Topic And Main Contributions:**

This paper proposes a fine-tuning method that updates the sub-net based on the parameter importance measured by the gradient of multiple dropout on the parameter. The effectiveness of the proposed method is verified on different datasets and different PLMs.

**Questions For The Authors:**

1. Can the authors explain some details about the ablation experiments? What is the impact of the two factors in 3.2.2?
2. Would the authors be willing to add more experimental results, such as experiments comparing with other methods on Roberta-Large?

**Reasons To Accept:**

1. According to the results presented by the authors, their method achieved good performance in different experiments.
2. In the robustness and generalization experiments, the proposed method shows more effective results.

**Reasons To Reject:**

1. The approach is not novel enough. Bi-Drop contains three parts, which are multiple dropout, parameter importance measure and dynamic update sub-net mask. In which all of them are used in the previous work. At the same time, the authors does not prove why to combine them through the experiments or theoretical analysis .
2. The experiment is not enough to prove the effectiveness of the method. The main experiments are only conducted on BERT-large, with little difference in performance. Meanwhile, the authors only compare with Vanilla method on other PLMs (e.g. Roberta).

**Reproducibility:**

3: Could reproduce the results with some difficulty. The settings of parameters are underspecified or subjectively determined; the training/evaluation data are not widely available.

**Reviewer Confidence:**

3: Pretty sure, but there's a chance I missed something. Although I have a good feel for this area in general, I did not carefully check the paper's details, e.g., the math, experimental design, or novelty.

---

> ### Author Rebuttal · Authors · 2023-08-29
>
> ***Q1: Can the authors explain some details about the ablation experiments? What is the impact of the two factors in 3.2.2?***
>
> R1: Thank you very much for your careful reading of our partial ablation experiment. As you mentioned, in Table 7 of the ablation experiment results, ESS represents our Effective Sub-net Selection strategy, which involves using the two factors described in section 3.2.2. In fact, we also conducted separate ablation experiments for these two factors, but due to space limitations, they were not included in the table. Now, we present a more detailed ablation result in the table below. Once again, thank you for your question.
>
> | **Method**                                | **CoLA**  | **MRPC**  | **RTE**   | **STS-B** | **Avg**   | **∆**     |
> | ----------------------------------------- | --------- | --------- | --------- | --------- | --------- | --------- |
> | **Bi-Drop** ( **g**$_{avg}$ + **ESS**)   | **64.94** | **91.79** | **73.79** | **90.59** | **80.26** | **+1.55** |
> | **g**$_{avg}$ **+ Perturbation Factor** | 64.71     | 91.67     | 73.21     | 90.36     | 79.99     | +1.28     |
> | **g**$_{avg}$ **+ Scaling Factor**      | 64.38     | 91.63     | 72.96     | 90.41     | 79.85     | +1.14     |
> | **g**$_{avg}$ **+** **RSS**             | 64.24     | 90.96     | 71.73     | 90.25     | 79.30     | +0.59     |
> | **g**$_{avg}$                           | 63.82     | 91.26     | 71.16     | 89.81     | 79.01     | +0.30     |
> | **Vanilla**                               | 63.76     | 90.41     | 70.97     | 89.70     | 78.71     | 0.00      |
>
> Caption: Ablation results. ESS represents our Effective Sub-net Selection strategy using both factors Perturbation and Scaling. RSS stands for Random Sub-net Selection strategy. Both our sub-net selection strategy and gradient averaging strategy are effective.
>
> ***Q2: Would the authors be willing to add more experimental results, such as experiments comparing with other methods on Roberta-Large?***
>
> R2: Based on your comments, we conducted experiments using the RoBERTa-large model. The specific results are shown in the table below.
>
> | **Method**            | **CoLA**         | **MRPC**         | **RTE**          | **STS-B**        | **Avg**          | **∆**           |
> | --------------------- | ---------------- | ---------------- | ---------------- | ---------------- | ---------------- | --------------- |
> | **Vanilla**           | 66.01(68.03)     | 92.56(93.66)     | 84.51(86.28)     | 92.05(92.22)     | 83.78(85.05)     | 0.00            |
> | **ChildTuning**$_D$ | 66.82(68.21)     | 92.67(93.58)     | 85.89(87.72)     | 92.36(92.53)     | 84.44(85.51)     | +0.66(0.46)     |
> | **DPS**$_{mix}$     | 66.86(68.53)     | 92.51(93.89)     | 85.37(87.72)     | 92.47(92.73)     | **84.30(85.72)** | +0.52(0.67)     |
> | **R-Drop**            | 67.26(69.63)     | 92.47(93.62)     | 85.44(87.63)     | 92.42(92.58)     | 84.40(85.87)     | +0.62(0.82)     |
> | **Bi-Drop**           | **68.03(70.89)** | **92.95(94.66)** | **86.10(88.09)** | **92.36(92.58)** | 84.86(86.56)     | **+1.08(1.51)** |
>
> Capiton: Comparison between Bi-Drop with prior fine-tuning methods. We report the mean (max) results of 10 random seeds. The best results are **bold**. Bi-Drop achieves the best performance compared with other methods.
>
> ***Q3: The approach is not novel enough. Bi-Drop contains three parts, which are multiple dropout, parameter importance measure and dynamic update sub-net mask. In which all of them are used in the previous work. At the same time, the authors does not prove why to combine them through the experiments or theoretical analysis .***
>
> R3:Thank you for your comment. Our work, similar to the CHILD-TUING and DPS mentioned in this paper, focuses on addressing how to effectively update sub-net in pre-trained models. Therefore, measuring parameter importance serves the purpose of facilitating sub-net updates, meaning that these two aspects are essentially the same. The key difference among different works lies in how the importance of different parameters is measured, as mentioned in the related work section of this paper. CHILD-TUING and DPS employ FI-based methods for this purpose. However, FI-based methods indeed rely heavily on the training data and can exhibit hysteresis in sub-net updating. In contrast, our method introduces an innovative approach by using different dropout settings to perform multiple forward propagations, resulting in multiple gradient values for each parameter. These gradient values are then utilized in the Perturbation Factor and Scaling Factor mentioned in the paper to calculate the importance of each parameter. Our method overcomes the limitations of the FI-based methods mentioned above, and we have demonstrated its effectiveness through extensive experimentation compared to FI-based methods. In summary, our innovation lies in proposing a new and effective strategy for measuring parameter importance (or, in other words, an effective subnetwork selection strategy), overcoming the limitations of FI-based methods. The paper provides a clearer demonstration of our method's specific workings, which is why it is divided into three parts in Section 3.2. However, in practice, it all boils down to performing effective subnetwork selection.

---

### Official Review · Reviewer_TJ6o · 2023-08-06

**Soundness:** 3

**Excitement:**

3: Ambivalent: It has merits (e.g., it reports state-of-the-art results, the idea is nice), but there are key weaknesses (e.g., it describes incremental work), and it can significantly benefit from another round of revision. However, I won't object to accepting it if my co-reviewers champion it.

**Missing References:**

See above.

**Paper Topic And Main Contributions:**

This paper improves the fine-tuning by updating partial parameters. This is achieved by learning a gradient mask based on dropout. It lies in the family of regularization methods. Experiments results on GLUE demonstrate its effectiveness.

**Questions For The Authors:**

See above.

**Reasons To Accept:**

1. Generalization in fine-tuning is an important problem
2. Masking part of the parameters makes intuitively sense because the LLM is known to be over-parameterization
3. The writing is clear


**Reasons To Reject:**

1. The idea of looking for important parameters and doing selective optimization is very similar to [1]. However, [1] uses soft-masking instead of hard-masking. What are the pros and cons compared to soft-masking methods?
2. Using fisher information is very straightforward when it comes to important computation. There is a lot of randomness in the dropout, how can you guarantee you are protecting what is important?
3. Can this method apply to parameter-efficient methods, like LoRA? Since this method physically changes the LM, it seems to me it is not applicable in this case. Also the authors may want to compare with those parameter-efficient method as well

[1]: Adapting a Language Model While Preserving its General Knowledge, Ke et al., EMNLP 2022

**Reproducibility:**

4: Could mostly reproduce the results, but there may be some variation because of sample variance or minor variations in their interpretation of the protocol or method.

**Reviewer Confidence:**

2: Willing to defend my evaluation, but it is fairly likely that I missed some details, didn't understand some central points, or can't be sure about the novelty of the work.

---

> ### Author Rebuttal · Authors · 2023-08-29
>
> ***Q1: The idea of looking for important parameters and doing selective optimization is very similar to [1]. However, [1] uses soft-masking instead of hard-masking. What are the pros and cons compared to soft-masking methods?***
>
> R1: Based on your statement, we carefully reviewed the paper [1]. Although our paper shares some similarities with the ideas presented in [1], there are significant differences between the two approaches. **Firstly**, the motivations behind [1] and our research are different. While [1] focuses on identifying important heads that capture pre-trained knowledge and applies a soft-mask to those heads, our motivation is to update parameters that are most useful for the current task while conservatively masking the remaining parameters. **Secondly**, in [1], a masking strategy is applied to each head of the multi-head attention, while in our method, we selectively mask individual parameters across the entire network. **Thirdly**, [1] utilizes a soft-mask, which requires introducing a learnable gate vector for each head of the attention mechanism. The gate vector's average gradient for each parameter is computed using the Kullback-Leibler (KL) divergence loss, and this average gradient is used as weights during backpropagation to achieve the soft-mask effect. On the other hand, our approach employs **Perturbation Factor** to select parameters that exhibit more stable gradients across multiple forward passes under different dropout configurations. This allows us to update these selected parameters without introducing additional learnable parameters, making our method simpler and more efficient. **Based on the three points mentioned above**, [1] only applies the masking strategy to a few heads of the multi-head attention, which enables them to introduce a relatively small number of learnable parameters for the soft-mask implementation. However, our method is designed for the entire model by selectively choosing sub-net based on rule-based strategies, without introducing any learnable parameters. If we were to implement a soft-mask approach, it would require introducing a learnable parameter for every parameter, which could lead to a significant increase in memory and computational resources. Therefore, the soft-mask approach is not suitable for our method.
>
> ***Q2: Using fisher information is very straightforward when it comes to important computation. There is a lot of randomness in the dropout, how can you guarantee you are protecting what is important?***
>
> R2: As mentioned in the related work section of this paper, FI-based methods indeed rely heavily on the training data and can exhibit hysteresis in sub-net updating. Furthermore, as mentioned in the response to question 1, the motivation behind this work is different from the motivation in paper [1]. Reference [1] focuses on identifying and preserving parameters that are crucial for the knowledge acquired during pre-training. In contrast, our approach involves selecting parameters that are important for the current task. We only update these important parameters, while the rest of the parameters remain unchanged. Therefore, from this perspective, our method is more conservative, as we protect parameters that are deemed less important for the current task.
>
> ***Q3: Can this method apply to parameter-efficient methods, like LoRA? Since this method physically changes the LM, it seems to me it is not applicable in this case. Also the authors may want to compare with those parameter-efficient method as well.***
>
> R3: Thank you very much for your advice. In fact, our approach and the parameter-efficient methods are orthogonal, so our approach can be applied on top of methods like LoRA. In the future, we can verify the effectiveness of applying our approach on top of methods like LoRA.
>
> [1]: Adapting a Language Model While Preserving its General Knowledge, Ke et al., EMNLP 2022

---

### Meta-Review · Area_Chair_G4ZT · 2023-09-13

**Recommendation:** 3

**Metareview:**

The experiments in the paper is sound. All reviewers agree the results support the claim about generalization in the paper.

However, there are a few concerns. Reviewer uYzG raises a valid concern that the improvement is not as significant. Reviewer TJ6o and NTTU raises concern on the applicability and computational cost of the approach. Overall, the authors should revise the paper to discuss whether the marginal improvement is worth the extra effort.

The writing needs some more work. There are several bogus claims and statements.

For example, as pointed out by Reviewer NTTU, in line 187,

> ... CHILD-TUNING_D only optimizes an unchanged sub-net during fine-tuning and ignores the update of other parameters, which may degrade the model’s performance on downstream tasks.

The sentence is too defensive that it's simple vacuous, but at the same time, the statement is likely blaming the wrong thing.

Another example is the problem stated in the Introduction:

> (1) hysteresis in sub-net updating: the sub-net preference is estimated with the model parameters in previous iterations and may be incompatible with the current update step; and (2) insufficient utility of training data: FI estimation requires cumulative gradients through multiple mini-batches, so these methods cannot fit in situations with data scarcity.

The description is imprecise, and the paper does not validate these problems before going about solving them. Again, we do not know if the paper is blaming the right things.

Overall, the paper has merit, but I would highly recommend the authors review and improve the wording in their revision.

---

### Decision · Program_Chairs · 2023-10-07

**Decision:**

Accept-Findings

**Comment:**

The experiments in the paper is sound. All reviewers agree the results support the claim about generalization in the paper.

However, there are a few concerns. Reviewer uYzG raises a valid concern that the improvement is not as significant. Reviewer TJ6o and NTTU raises concern on the applicability and computational cost of the approach. Overall, the authors should revise the paper to discuss whether the marginal improvement is worth the extra effort.

The writing needs some more work. There are several bogus claims and statements.

For example, as pointed out by Reviewer NTTU, in line 187,

> ... CHILD-TUNING_D only optimizes an unchanged sub-net during fine-tuning and ignores the update of other parameters, which may degrade the model’s performance on downstream tasks.

The sentence is too defensive that it's simple vacuous, but at the same time, the statement is likely blaming the wrong thing.

Another example is the problem stated in the Introduction:

> (1) hysteresis in sub-net updating: the sub-net preference is estimated with the model parameters in previous iterations and may be incompatible with the current update step; and (2) insufficient utility of training data: FI estimation requires cumulative gradients through multiple mini-batches, so these methods cannot fit in situations with data scarcity.

The description is imprecise, and the paper does not validate these problems before going about solving them. Again, we do not know if the paper is blaming the right things.

Overall, the paper has merit, but I would highly recommend the authors review and improve the wording in their revision.